# RE-AIM evaluation of a community-based vaccine education and communication program to improve human papillomavirus vaccine uptake in Tonga

Yasmin Mohamed[1,2]*, Isabella Overmars[1], Ofakiokalani Tukia[3], Emma Luey[4], 'Asinate Toluta'u[4], Meleane Lomu[4], Luisa Vodonaivalu[1], Afu Tei[3], Reynold 'Ofanoa[3], 'Ungatea Kata[4], Julie Leask[5], Holly Seale[6], Kylie Jenkins[1], Kshitij Joshi[7], Halitesh Datt[7], Sonya Sagan[7], Michelle Dynes[8], Jessica Kaufman[1,2], Margie Danchin[1,2,9]

**1** Murdoch Children's Research Institute, Parkville, Victoria, Australia, **2** Department of Paediatrics, The University of Melbourne, Parkville, Victoria, Australia, **3** Tonga Ministry of Health, Nuku'alofa, Tonga, **4** Tupou Tertiary Institute, Nuku'alofa, Tonga, **5** School of Public Health, The University of Sydney, Camperdown, New South Wales, Australia, **6** School of Population Health, The University of New South Wales, Kensington, New South Wales, Australia, **7** UNICEF Pacific, Suva, Fiji, **8** UNICEF East Asia and Pacific, Bangkok, Thailand, **9** The Royal Children's Hospital, Parkville, Victoria, Australia

* ymohamed@student.unimelb.edu.au

## Abstract

There is a global need for effective strategies to improve acceptance and uptake of the human papillomavirus (HPV) vaccine, particularly in the Pacific where cervical screening and treatment options are limited. While the Pacific Island nation of Tonga has high routine childhood vaccine coverage, HPV vaccine uptake has remained low since its introduction in 2022. To improve vaccine acceptance and uptake, recent evidence supports the use of community engagement approaches like the Vaccine Champions program, which trains health and community leaders to advocate for vaccines. We assessed the reach, effectiveness, adoption, implementation, and maintenance of the Vaccine Champions program in Tonga. In March 2023 we conducted a co-design workshop with government and non-government stakeholders to adapt the program and establish key features. Diverse Vaccine Champions were trained in June to run vaccine information sessions in their communities until December 2023. We used the RE-AIM framework to evaluate program implementation and impact through surveys and interviews with co-design participants, Vaccine Champions, and community members. Co-design participants (n = 29) agreed to focus on the HPV vaccine, identified leading barriers and potential Vaccine Champions. We trained 27 Vaccine Champions including teachers, local officials, nurses, and representatives from disability, youth, and sports groups. Most were female (19/27; 70%) and had no health background (25/27; 93%). Training increased Vaccine Champions' trust and support for vaccines, and confidence to communicate about vaccines. Vaccine Champions ran 57 vaccine information sessions, reaching 1138 community

**Data availability statement:** Our dDe-identified quantitativedataset for the quantitative data is data are now available through the link we shared in our submissionat: ([https://doi.org/10.25374/MCRI.c.7902611](https://doi.org/10.25374/MCRI.c.7902611)) [https://doi.org/10.25374/MCRI.c.7902611](https://doi.org/10.25374/MCRI.c.7902611). Qualitative data are available from the corresponding author upon reasonable request.

**Funding:** This work was supported by the Australian Government under the Australian Regional Immunisation Alliance - Regional Immunisation Support and Engagement grant. This research was completed as part of a PhD funded by a National Health and Medical Research Council of Australia scholarship (2022510) and the Australian Government's Research Training Program. Neither funding body had any role in the study design, data collection, analysis and interpretation, or the decision to submit this manuscript for publication.

**Competing interests:** OT, AT, and RO are employed by the Tonga Ministry of Health and involved in the national HPV vaccine program. HS has received funding from vaccine manufacturers for investigator-driven research and has consulted on COVID-19 vaccination for Pfizer. This funding was not related to this work. All other authors declare no competing interests. There are no patents, products in development, or marketed products associated with this research to declare. This does not alter our adherence to PLOS ONE policies on sharing data and materials.

members. Parent or caregiver intention to vaccinate their daughters against HPV increased from 70% to 88% after attending a community session. The Vaccine Champions program is a culturally tailored community engagement approach to increase HPV vaccine acceptance and uptake, supporting global efforts towards cervical cancer elimination. It is continuing in Tonga in 2025 with strong national support. Implementation and evaluation process findings can support program localisation elsewhere.

## Introduction

Over 340,000 people die from cervical cancer each year, most in low- and middle-income countries [1]. There is a global drive to eliminate cervical cancer through vaccination against the human papillomavirus (HPV), cervical screening, and treatment [2]. All currently available HPV vaccines are highly effective against the two virus types causing over 70% of cervical cancer cases, and additionally protect against many other forms of cancer [3]. A single dose schedule has been shown to be effective [4], yet globally only 27% of eligible girls are vaccinated against HPV [5]. Unlike early childhood vaccines, the HPV vaccine is generally delivered to adolescents in schools, which can present logistical challenges. Persistent vaccine concerns and misinformation also affect acceptance [6].

The Kingdom of Tonga is an upper-middle income Pacific Island Country [7] comprising 176 islands [8]. Most of the population of 100,000 live on Tongatapu Island [8,9]. Full immunisation coverage for children aged 12–23 months is 93.5% [10], and there are high levels of trust and confidence in the childhood vaccination program [11]. In November 2022, Tonga introduced the bivalent HPV vaccine [12]. A free one-dose schedule is integrated into the national immunisation schedule for 10-year-old girls and available for girls aged 11–14 years through a catch-up campaign. Prior to vaccine introduction, a survey of caregivers in Tonga indicated that only 37% had heard of the HPV vaccine [11]. After learning about it through the survey, 92% stated they intended to vaccinate their child against HPV [11], but by February 2023 the Ministry of Health estimated that only 20% of eligible girls had received the vaccine [13], leading them to seek strategies to bridge the intention-uptake gap.

While most caregivers in Tonga get information about vaccines from health workers, over one third (35%) report friends, family, and neighbours as their main source of information [11]. Engaging trusted members of the community, including those outside the health sector, can increase vaccine confidence and trust and improve vaccine uptake [14]. Interventions that engage communities to build knowledge and trust in vaccines have been used globally to increase uptake of routine childhood and adult COVID-19 vaccines [15,16], and community leaders have an important role in communication activities around HPV vaccine implementation [6]. A recent systematic review of behavioural interventions to increase vaccination rates found that engaging vaccine champions to encourage vaccination was an effective strategy, provided they received adequate training and support [17]. Systematic reviews also support the use of multi-level interventions to increase HPV vaccine uptake [18].

This study aimed to evaluate the implementation and impact of the Vaccine Champions program, a comprehensive vaccine education and communication skills training program for diverse community advocates to improve acceptance and uptake of the HPV vaccine in Tonga. The program targets multiple drivers of vaccine uptake by improving knowledge, addressing community concerns and misinformation, capitalising on social norms, and reminding people about vaccination [14,19]. We used the RE-AIM framework to measure and describe reach, effectiveness, adoption, implementation, and maintenance [20].

## Methods

This paper presents the results of a mixed methods evaluation of the Vaccine Champions program, focusing on how each phase of the program was implemented and its outcomes. The next section provides an overview of the three phases of the program itself, then moves on to details of the evaluation methods.

### Program overview

The Vaccine Champions program is a community engagement approach that trains healthcare workers and community leaders to become effective vaccine advocates, providing them with the knowledge, communication skills, and required resources to run vaccine information sessions in their communities [14,19]. The three program phases are outlined in Fig 1. It was initially developed in Australia to facilitate COVID-19 vaccine uptake [19], and has since been implemented and evaluated in Fiji [14] and Viet Nam with a focus on COVID-19 and routine childhood vaccines. In 2023 in partnership with the Tonga Ministry of Health, UNICEF Pacific, and the Tupou Tertiary Institute, we adapted, implemented, and evaluated the program in Tonga. This was the first time the program had focused on the HPV vaccine.

Phase 1 was a half-day co-design workshop to adapt the program for Tonga. The Ministry of Health invited representatives from government, non-government organisations, and community groups. An overview of routine childhood and HPV vaccine coverage rates in Tonga was followed by an activity identifying and mapping barriers and drivers of HPV vaccine uptake using the World Health Organization's (WHO) Behavioural and Social Drivers of vaccination (BeSD) framework [21]. Participants then voted on program features using dot voting to prioritise from a range of options. The core components of the program were predetermined, however the features decided at the co-design workshop included vaccine focus, who should be a Vaccine Champion, how to incentivise Champions, and length of Champions' training. The study team or Ministry of Health invited individuals or organisations to attend the Vaccine Champions training.

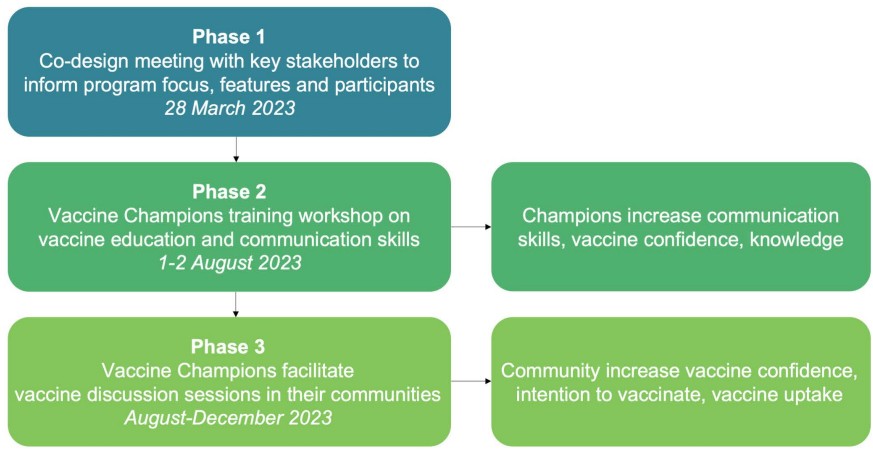

**Fig 1. Outline of the Vaccine Champions program in Tonga.**

Phase 2 was a two-day training workshop in Tongatapu to train Vaccine Champions, delivered in English with local staff from Tupou Tertiary Institute and the Ministry of Health translating as required. Training materials were available in Tongan and English and were provided online for a participant with a visual impairment to process with appropriate software. Training topics included the HPV vaccine, routine childhood vaccines, vaccine communication skills including conversations with vaccine hesitant people, addressing misinformation, and practical tips on hosting community information sessions. Champions practiced communication skills through structured role plays [14,22]. They were given some suggestions for what to cover in a community information session and were encouraged to use their vaccine knowledge and communication skills to share information about the HPV vaccine with their community in any way that worked for them. To support their community information sessions, Champions received Tongan language materials including copies of the training presentations, an activity workbook, a guide on running a community session, and a program t-shirt.

In Phase 3, Vaccine Champions were asked to run up to three vaccine information sessions. Champions were encouraged to tailor the sessions, which could include speaking at an existing gathering; organising a specific meeting to talk about vaccines; or opportunistically discussing vaccination with individuals. The community members who attended a Vaccine Champion's information session were included as "community attendees" in the program evaluation. The Tongan research team regularly contacted Champions by phone or Facebook Messenger to encourage them to run sessions, answer questions, and provide logistical support. Champions received a phone card for 20 Tongan Pa'anga (around USD$8.50) for each session they ran.

## Evaluation design

We evaluated program implementation and impact using the RE-AIM framework, assessing intervention reach, effectiveness, adoption, implementation, and maintenance [14,20]. This framework is commonly used to evaluate public health and behaviour change interventions [23]. It enables understanding of individual and community-level outcomes, as well as exploring how and why the intervention works [20]. The RE-AIM domains measured for each program phase are outlined in Table 1.

**Table 1. Data collection methods and outcomes for the Vaccine Champions program evaluation in Tonga.**

| Program phase | Participant group | Data collection methods | RE-AIM domain and outcomes |
|---|---|---|---|
| Phase 1 | Co-design participants | • Post co-design survey<br>• End of program interview<br>• Research team field notes | **Reach:** number and characteristics of co-design attendees<br>**Effectiveness:** satisfaction with co-design and overall program<br>**Implementation:** overall program implemented as intended<br>**Maintenance:** perceived sustainability of program features |
| Phase 2 | Vaccine Champions | • Pre-training survey<br>• Post-training survey<br>• Reflection forms<br>• End of program interview<br>• Research team field notes | **Reach:** number and characteristics of Champions trained<br>**Effectiveness:** changes in vaccine knowledge, confidence, and trust<br>**Effectiveness:** satisfaction with the training and overall program<br>**Adoption:** number of Champions who delivered info sessions<br>**Adoption:** characteristics of Champions who didn't run sessions<br>**Implementation:** overall program implemented as intended<br>**Maintenance:** willingness to continue program<br>**Maintenance:** perceived sustainability of program features |
| Phase 3 | Community members attending Vaccine Champion sessions | • Post-session survey<br>• Champion reflection forms<br>• Post-session interview<br>• Research team field notes | **Reach:** number and characteristics of community members attending info sessions<br>**Effectiveness:** changes in vaccine confidence and trust<br>**Effectiveness:** change in intention to vaccinate daughter<br>**Effectiveness:** satisfaction with Champions' sessions<br>**Maintenance:** willingness to continue program |

## Participants and data collection

We evaluated the three program phases using qualitative and quantitative methods (Table 1). Study participants were co-design participants (Phase 1), Vaccine Champions (Phase 2), and community members attending Champions' sessions (Phase 3). Recruitment started at the co-design on 28 March 2023 and finished on 31 December 2023.

### Phase 1: Co-design participants

Co-design participants were invited to complete a survey in English immediately after participating in the co-design workshop. Interviews with co-design participants involved in program implementation were conducted in English by the first author.

### Phase 2: Vaccine Champions

Vaccine Champions completed surveys before and immediately after training, measuring knowledge, vaccine confidence and trust, and communication self-efficacy, as in our Fiji program evaluation [14]. Knowledge questions were developed by the research team. Vaccine confidence and trust were measured through seven items adapted from the WHO tools for understanding the behavioural and social drivers of vaccination [21]. We measured communication self-efficacy with four questions adapted from Bandura's self-efficacy scale [24].

Vaccine Champions completed a reflection form after each community session, capturing their experiences and personal reflections, and recording the number (collected as a range in increments of five) and type of community members invited to and attending the session. Champions measured attendees' intention to get the HPV vaccine for their daughters with a show of hands before and after each session. All data collection forms for Champions were available in Tongan and interviews were conducted by the local research team in Tongan or English.

### Phase 3: Community attendees

Community attendees (people attending a vaccine information session run by a Vaccine Champion) completed a short paper survey in Tongan post-session, collecting demographic details and satisfaction. Self-reported changes in vaccine confidence and trust were measured with four questions adapted from the WHO BeSD tools [21], using a 3-point scale from "less than before" to "more than before". Community attendees were also asked if they would like to take part in an interview. If they did, they had the option of adding their contact details to the Champions' reflection form and were later contacted and invited to take part in a qualitative interview in Tongan, conducted by Tongan members of the research team.

Qualitative question guides for all interviews were adapted from the WHO BeSD tools [21] and drew on Ripple Effect Mapping methods, a participatory approach for evaluating complex community-based programs [25].

Survey and interview data from co-design participants were collected in English by the first author. Data collection with Vaccine Champions and community attendees was conducted by Tongan members of the research team in Tongan or English.

### Sampling

All co-design participants, Vaccine Champions, and community attendees were invited to complete surveys. A purposive sample of community attendees were invited to be interviewed. All Champions and a purposive sample of co-design participants involved in program implementation were invited to take part in an interview at the program's end.

### Ethical approval

Ethical approval was obtained from the Royal Children's Hospital Human Research Ethics Committee (HREC: 84863), and the Tonga National Health Ethics and Research Committee (MH 53:02). All participants provided written consent (for

co-design participants and Vaccine Champions), verbal informed consent (for community attendee interviews), or implied consent (for community attendee surveys). Community surveys were anonymous, optional and did not collect any identifying data; surveys had a Participant Information Statement and completion of the survey was considered as consent to participate. In the case of verbal consent, the local research team provided community attendees with information outlining what the study was about, how their responses would be kept confidential, and what the data would be used for. All consent processes were finalised in collaboration with our local research partners and the Ministry of Health and approved by the ethics committees in Tonga and Australia.

### Data analysis

Quantitative categorical responses are presented as numbers and percentages at each timepoint. Knowledge is reported as the percentage of participants who correctly identified misinformation about vaccines, recommended the correct communication techniques, and correct information on childhood vaccines. Trust and confidence in vaccines is reported as the percentage of participants responding "very much", and communication self-efficacy as those who responded "very confident". McNemar's chi-squared test was done on the pre-post responses, with responses collapsed to binary variables. Training satisfaction is reported as those responding "very satisfied". The number of community attendees at each session was completed as a range; the mean of the upper and lower estimates was used to estimate the average, range, and total number of attendees. We used STATA statistical software version 18.0 for analysis.

Qualitative data were collected to supplement the quantitative findings and provide additional in-depth insights [26]. Qualitative data (transcripts, reflection forms and field notes) were transcribed and translated where necessary, then analysed by the first author with NVivo version 12, using a deductive content approach [27]. The research team discussed and agreed upon the coding framework for analysis, then mapped the codes to the different dimensions of the RE-AIM framework. Here we present quotes that are illustrative of common perspectives from across the qualitative findings as well as individual divergent perspectives.

### Patient and public involvement

Community members were involved in the design and the delivery of the intervention. Members of local community groups attended the co-design workshop to discuss and identify key barriers and facilitators to immunisation and to inform the design of the intervention. Trusted community members who are leaders in their communities were involved in the delivery of the intervention as trained Vaccine Champions.

## Results

Table 2 provides an overview of evaluation participants at all levels. Findings are detailed below.

### Co-design workshop

The results of the voting by co-design participants were a program focused on the HPV vaccine; a two-day training workshop for Vaccine Champions; a more diverse range of Champions with smaller numbers in each group; and most Champions to come from the main island of Tongatapu.

### Reach

Twenty-nine of 37 invited stakeholders (78%) attended the co-design workshop in March 2023 with most being female (26/29; 90%). Stakeholders included representatives from the Ministries of Health, Education and Training, and Internal Affairs; UNICEF; Tonga Red Cross; health workers; university and tertiary institute staff; and representatives from faith organisations, sporting groups, a women's empowerment organisation, and the national youth congress. Those who did

**Table 2. Evaluation participants for the Vaccine Champions program in Tonga.**

| Participant group | Program activity | Number of activities | Reach – attended/invited | Surveys completed | Interviews completed |
|---|---|---|---|---|---|
| Co-design participants | Co-design workshop | 1 | 29/37 (78%) | 21 | 3 |
| Vaccine Champions | Vaccine Champions training workshop | 1 | 27*/45 (60%) | 25 (pre-training) 21 (post-training) 23 (reflection forms) | 11 |
| Community attendees | Community information sessions | 57 | 1138** (mean per session = 22) | 269 | 7 |

* Attended both days of the training

** Total number attending the sessions, collected as ranges in increments of five. Number invited was not collected.

not attend included representatives from non-government disability organisations, family health organisations, and the rugby association.

## Effectiveness

After the workshop, 21/29 (72%) participants completed a short evaluation survey. Most surveyed (18/21; 86%) were satisfied with the workshop; everyone understood the purpose of the day and felt they were able to share their perspectives. Feedback on the session from surveys and interviews was generally positive: "It was really very interactive, and even [had] new ways of how to get the key stakeholders engaged." (Key informant 2)

While most co-design participants were satisfied with the length, two survey participants felt the session could have been longer, to include more small group discussions. In an end of program interview, one co-design participant from the Ministry of Health felt it would have been beneficial to have the most senior staff from the Ministry attend the entire workshop to maximise high-level program support. Another interviewee thought there was good representation from health and non-health stakeholders.

An academic partnership between MCRI and the Tupou Tertiary Institute in Tonga arose from the co-design workshop.

## Vaccine champions training

**Reach.** Twenty-seven of the 45 invited community leaders (60%) attended both days of the workshop, with another three attending only the first day for unspecified reasons. Invited Champions who did not attend included representatives from local and national government, local non-government organisations, media, and family health organisations. Of the Champions who did attend both days of the training, most were female (19/27; 70%) and two were nurses (2/27; 7%; Table 3). Champions included teachers; school nurses; local government officials; staff from non-government organisations; representatives from disabled people's organisations; and members of youth, sports, and faith groups.

**Effectiveness.** Twenty Vaccine Champions (20/27; 74%) completed matched pre- and post-training surveys. Knowledge of vaccines increased after the training, and the proportion who identified appropriate communication techniques decreased (Fig 2). Trust and confidence in HPV and routine vaccines were high among the Champions and increased after the training (Fig 3). Vaccine Champions reported their confidence to communicate about vaccines increased after the training in all four areas assessed (Fig 4). This was also reflected in the interviews, with Champions describing how the training gave them confidence to speak about sensitive issues in front of a group. There was a statistically significant increase in the proportion of Champions who felt very confident to talk about vaccine side effects (pre-training 13/20; 65%; post-training 19/20; 95%; p = 0.031). None of the other changes were statistically significant (S1 Table).

**Table 3. Characteristics of Vaccine Champions (N = 27).**

| Characteristic | Number (%) |
|---|---|
| Island group | |
| Tongatapu | 21 (78) |
| Vava'u | 3 (11) |
| Ha'apai | 3 (11) |
| Gender | |
| Female | 19 (70) |
| Male | 8 (30) |
| Health background (nurse) | |
| Yes | 2 (7) |
| No | 25 (93) |
| Highest education level | |
| Primary | 2 (7) |
| Secondary | 7 (26) |
| Certificate/ TVAT | 9 (33) |
| University | 7 (26) |
| Missing | 2 (7) |
| Previous communication training | |
| Yes | 9 (33) |
| No | 16 (59) |
| Missing | 2 (7) |

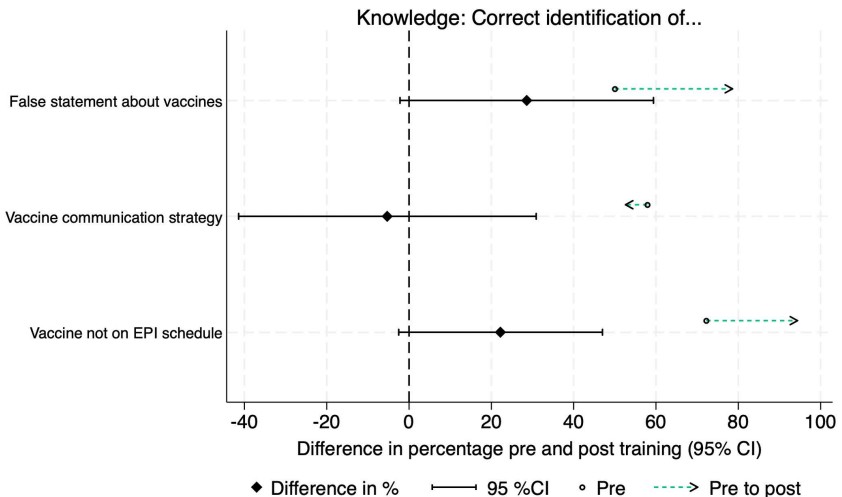

**Fig 2. Differences in vaccine knowledge pre- and post-training.**

Most Champions were very satisfied with the training (19/21; 90%) and the materials provided to run sessions (20/21; 95%). Champions also spoke highly about the training in interviews, highlighting that it was informative, included the right people and addressed key community concerns. One Champion felt it would have been better to only include women in the training given the sensitive nature of the HPV vaccine and another thought the training could have been longer.

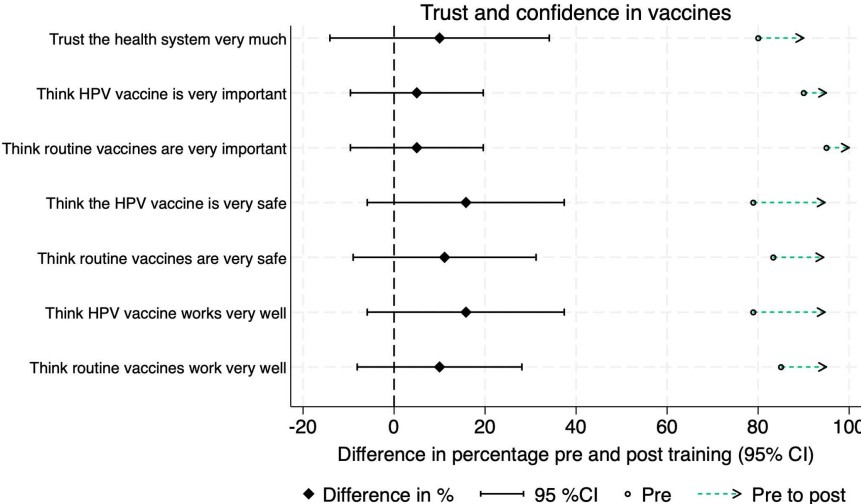

**Fig 3. Differences in vaccine trust and confidence pre- and post-training.**

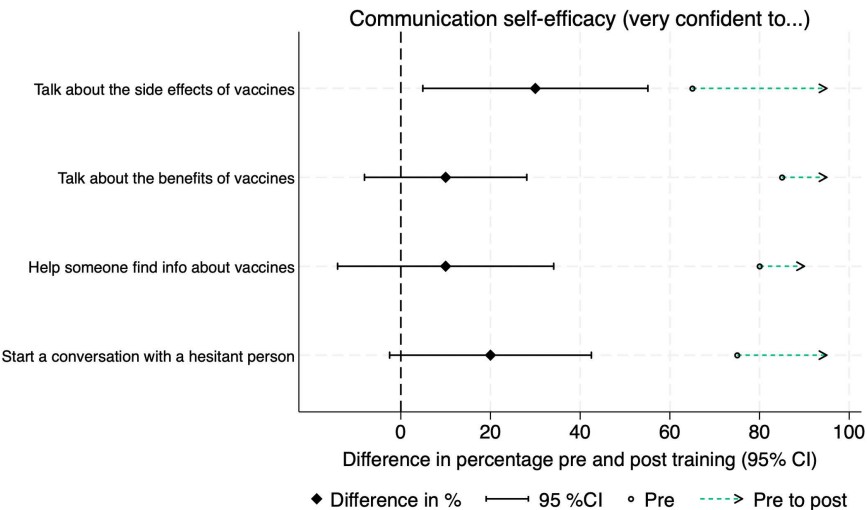

**Fig 4. Differences in communication self-efficacy pre- and post-training.**

One co-design participant felt the training should have been longer, and that the Tongan translation of materials could be improved. Conversely, a Vaccine Champion thought the translated information was "communicated simply and honestly" (Champion 4).

**Adoption.** Most Vaccine Champions (23/27; 85%) ran at least one vaccine information session, and 44% (12/27) ran three or more, with four Champions running five sessions. Interviewed Champions spoke about their motivations for becoming a Vaccine Champion. Many described wanting to help their communities, especially the children and young people. Two spoke specifically about a desire to protect their own children: "as mothers we want our children to grow up healthier than us who were not able to have this opportunity" (Champion 24). Champions felt that having the support of local government and church leaders facilitated organising and running sessions.

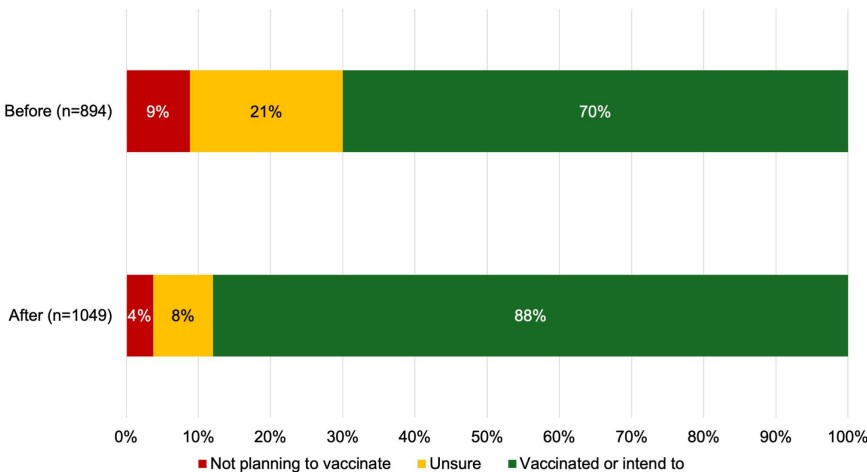

Global Public Health

We interviewed several Champions who did not run multiple sessions. The barriers they highlighted to running one or more sessions included lack of time, low interest from parents, and not being able to obtain approval from school principals to run sessions with students. One Champion (teacher) left the program after running one session because of their own personal beliefs about the HPV vaccine.

"I just withdrew from this [program] cause I know I don't have the same beliefs with what was being done. Because I didn't believe in the vaccine." (Champion 3)

Another Vaccine Champion chose not to run any more sessions because they felt discouraged after an attendee publicly shared some misinformation about the HPV vaccine during their first session. This male Champion also reported feeling uncomfortable speaking about the HPV vaccine in front of women.

"Women's [issues or bodies] was brought up, that made me feel embarrassed to hear discussions about that. […] the truth is it is better for only women to talk about those things." (Champion 18)

### Community sessions

**Reach.** Vaccine Champions ran a total of 57 community information sessions, reaching an estimated 1138 people. Sessions had a mean of 22 attendees, with a range of less than five people to 55 community members present. Attendees included parents, teachers, students, members of sports or church groups, and people living with disability. Almost one quarter of attendees completed a post-session survey (24%; 269/1138); most survey participants were female (209/269; 78%) and resided in Tongatapu (183/269; 69%). Community members who completed surveys attended sessions run by 19 different Champions, with only four Champions who ran sessions having no surveys completed about at least one of their sessions. There were survey participants from all island groups within Tonga including one participant from the Niuas (S2 Table).

**Effectiveness.** The proportion of community members who planned to get their child vaccinated against HPV increased from 70% to 88% after attending a Vaccine Champion session (Fig 5).

Three quarters of community members who completed the survey (199/269; 74%) reported being satisfied with the session. Community attendees reported more confidence in the importance of vaccines and being more likely to recommend the HPV vaccine to caregivers of eligible girls after attending the session (Fig 6).

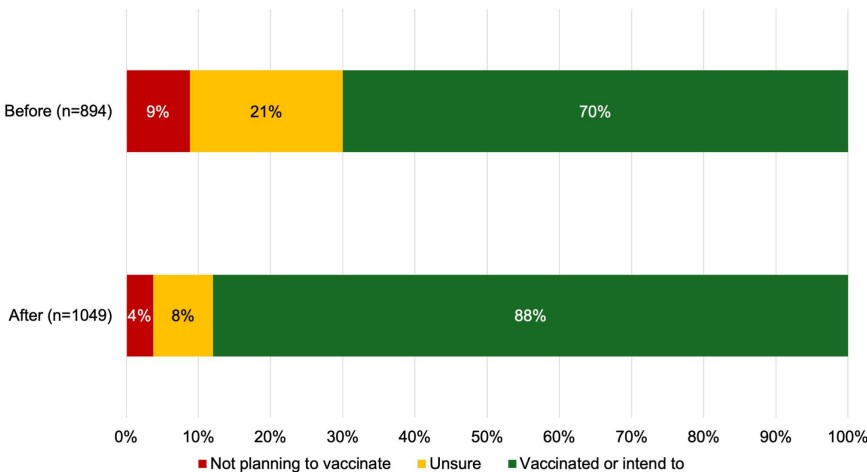

**Fig 5. Community attendees' intention to vaccinate their child against HPV before and after a Vaccine Champion information session.**

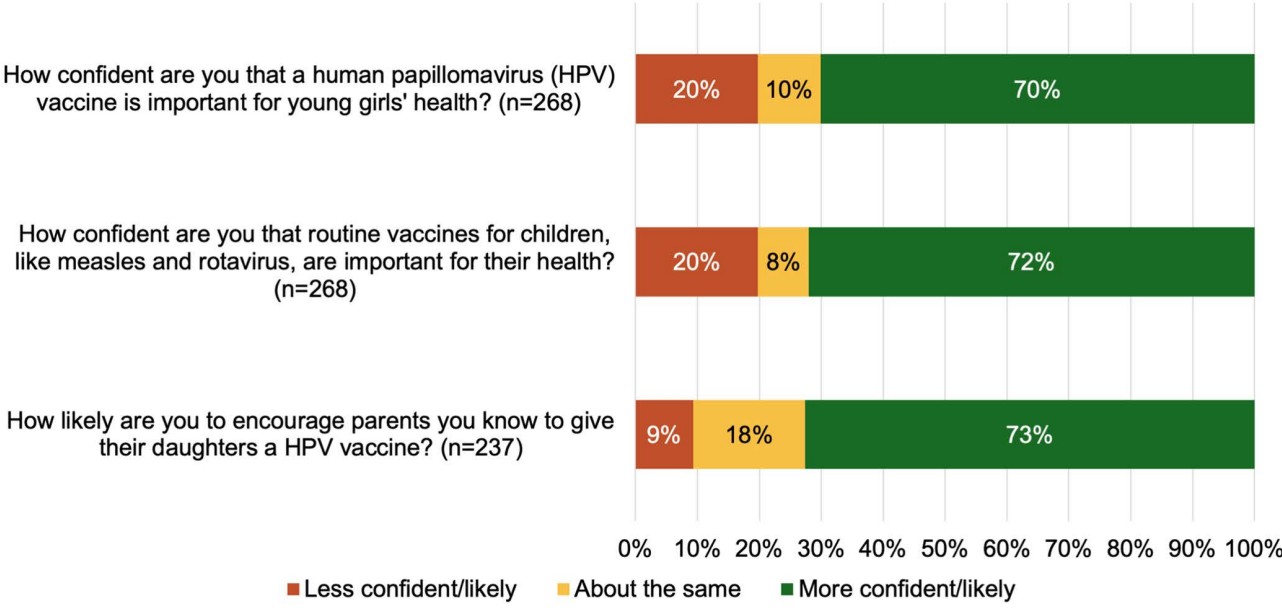

**Fig 6. Community attendees' support for vaccines after attending a Vaccine Champion information session.**

Vaccine Champions spoke about the benefits of their sessions, including connecting more with their communities. For example, a Champion working as a local government officer described members of their community choosing to get the HPV vaccine for their daughters after listening to their information sessions.

"I talked to the town officers for feedback, and they said that a higher percentage of those were vaccinated after we did the sessions." (Champion 15)

**Adoption.** Of those who completed the community attendee survey, 68% were more likely to start a conversation about vaccines with others after attending a Vaccine Champion session (179/263). We interviewed seven community members who attended one Vaccine Champion's session. All spoke positively about the session and reported sharing information about the HPV vaccine with family members and friends afterwards.

"I talked to my aunt about [the HPV vaccine] and she said she heard about it before and now that she confirmed it is good she is going to take her two daughters to be vaccinated, and they did the following day." (Attendee 6)

More specifically, one attendee said she would now encourage her younger sister to get the HPV vaccine after hearing about the benefits. Another described listening to her cousin's concerns about the vaccine, answering her questions and then explaining the vaccine's purpose and importance.

## Overall program

**Implementation.** The HPV vaccine was already available in Tonga during program implementation, and an awareness campaign was run by the Ministry of Health in August 2023. This campaign comprised communication activities including radio and television promotion and nationwide training for nurses. In Tonga, HPV vaccine coverage increased from 20% in February 2023 to 32% in October 2023 according to routinely collected national data [13,28].

Vaccine Champions customised their community information sessions. Most were in front of small groups and focused on the HPV or routine vaccines for children. One Champion chose to do phone calls with individual parents as this fitted better with their role and availability.

Champions described challenges with running sessions including a lack of vaccine confidence and trust in the community since the COVID-19 vaccine rollout; concerns about side effects of the HPV vaccine; community expectations that health information should be shared by trained health workers; a lack of materials to support their session; and not enough time to share their message. One Champion would have liked pamphlets to run their sessions; another thought the materials provided were helpful in supporting vaccine conversations.

Co-design participants and Vaccine Champions had suggestions for improving program implementation. Some thought that having Champions with fulltime jobs who could only run sessions out of work hours made it difficult for them to link with nurses providing the HPV vaccine. They suggested including people who have more time during the day such as pastor's wives, stay-at-home parents, or church ministers. Conversely, another Champion thought that the right people had been chosen as many of them were mothers and had "the right attitude" (Champion 12). While one Champion felt that men were not the right people to talk about a vaccine to prevent cervical cancer, a co-design participant felt it was a strength to have male Champions.

**Maintenance.** All co-design participants interviewed felt the program should be continued. Most interviewed Champions wanted to remain involved and share information about the HPV vaccine with their communities "because it benefits our women." (Champion 15). Community attendees also wanted to attend more sessions in the future.

To support sustainability, Champions and co-design participants thought some form of incentive was needed for the Champions. This could be a monetary payment or reimbursement for the cost of running a session such as travel, refreshments, and venue hire. Culturally appropriate materials to support the sessions such as pamphlets or flip charts were also seen as important.

Co-design participants and Champions described the importance of developing stronger links between Champions and nurses if the program were to continue. This could include involving nurses in choosing the Champions.

"It is a very good program because nurses need support in the community. But when we identify the [right] Champion, it will be much better. It will strengthen the working in partnership with the community." (Co-design participant 1)

## Discussion

Our Vaccine Champions program trained 27 diverse health and non-health Champions in HPV vaccine knowledge and communication skills, reaching over one thousand community members across the four main island groups in Tonga. Most trained Champions ran community sessions and remained engaged in the program, despite not having a health background. Training improved Champions' vaccine knowledge and confidence to communicate about vaccines. Champions' sessions increased community members' intention to vaccinate their daughters against HPV. Co-design participants believed the program was beneficial for supporting culturally appropriate HPV vaccination in Tonga. The majority of Vaccine Champions and co-design participants wanted the program to continue, with suggested adaptations. Given the positive response and success of the program in Tonga, the Ministry of Health is extending the program in 2025 using a train-the-trainer model to maximise reach, local ownership and long-term sustainability.

The Vaccine Champions program engages community leaders in design and implementation. A recent systematic review of community engagement interventions in low- and middle-income countries found a small but significant positive effect on childhood immunisation coverage and timeliness [29]. In addition to health providers, the role of trusted community members in improving vaccine acceptance and uptake is increasingly recognised, particularly since the COVID-19 pandemic [30–33] and was demonstrated in our Vaccine Champions program in Fiji [14]. One Champion chose to leave

the program, stating that the HPV vaccine did not fit with their beliefs. Outlining clear selection criteria for future Champions could mitigate this unintended consequence, ensuring that people who support vaccination are chosen.

Our program included male and female Vaccine Champions. While some female Champions and co-design participants saw this as a strength of the program, one male Champion did not feel that it was his place to speak about women's reproductive health. In Tongan culture, "anga faka'apa'apa" or respectful behaviour includes not discussing reproductive health issues with both males and females together [34]. This cultural concept may also mean that the male Champion was not able to speak freely when interviewed by a female member of the research team. Ensuring that male Champions only speak with groups of men and including men in the research team could have helped address this challenge. A global move towards gender-neutral HPV vaccination suggests that men and boys do have a role to play in promoting the HPV vaccine [35]. While mothers tend to be the primary decision-maker about childhood immunisations in Tonga, fathers do still have an important role to play in vaccine decision-making and sharing information with other fathers [11,36]. Whether or not male Champions are included in the program in future should be carefully considered.

Similar to other studies, Champions' knowledge and confidence to communicate about vaccines increased post- training, with a statistically significant increase in Champions who felt very confident to talk about vaccine side effects [37,38]. There was a non-significant reduction in the proportion of Champions able to correctly identify an effective communication technique from a list of options. Given our small sample size, this corresponds to one additional Champion answering the question incorrectly. One Champion was not confident to respond to misinformation from a session attendee. This suggests the need for training improvements such as incorporating more experiential learning techniques [39], longer sessions, more emphasis on role-playing and practicing skills, or including prereading materials. Incentives for Vaccine Champions should also be considered. While financial incentives can be challenging to maintain, appropriately compensating community volunteers is important to maintain their motivation and engagement [40]. The continuation of the Vaccine Champions program in Tonga in 2025 includes monetary incentives for Champions.

Community intention to vaccinate their daughters against HPV increased after attending a Vaccine Champion session, and satisfaction was high. Other studies in the region also show that higher parental knowledge and satisfaction with information correlate with increased intention to vaccinate [41,42]. Our community survey results show increased support for vaccines after attending a Vaccine Champion session, highlighting the importance of social mobilisation for increasing HPV vaccine acceptability [43]. A similar program in Nigeria that trained volunteer community mobilisers also found an improvement in attitudes towards immunisations [44]. However, one fifth of our community survey participants reported being less confident about the HPV and routine childhood vaccines after attending a session. This could be due to translation challenges, or to attendees becoming more aware of the HPV vaccine and having additional questions. Having a nurse support the Vaccine Champion sessions, or encouraging follow up between Champions and community members, could mitigate this unintended outcome.

## Strengths and limitations

Mixed methods at all levels of data collection allowed for data triangulation and strengthened the validity of our findings. Structuring our data collection around the RE-AIM framework improved the quality and comparability of our evaluation [45]. One key strength essential for successful implementation was engagement of a local research team from Tupou Tertiary Institute in Tonga, who maintained regular contact with the Champions. The study also had strong support from the Tonga Ministry of Health and UNICEF, essential to implementation and long-term sustainability.

There were several limitations. Electronic data collection methods were not possible due to variable internet and paper-based surveys were not always completed. While interviews provided more depth, a limited number of community attendees were able to be recruited from Vaccine Champion sessions due to resource constraints. We were also unable to contact three of the four trained Champions who did not run any sessions and were unable to explore their reasons.

Our translated surveys were reviewed by the local research team to maximise understanding but self-reported measures of confidence and satisfaction may be impacted by social desirability bias. The combination of response bias and social desirability bias may mean that those who completed the surveys had a more positive view of the program than those who did not. This is particularly true for the community surveys, where only 24% of attendees completed a survey. All Champions were asked to distribute surveys at their sessions and survey participants attended sessions run by most Champions. However, it is possible that those whose sessions went well were more likely to hand out the survey to community attendees. Our overall sample size for the surveys is small, limiting the statistical power of our analysis.

Effectiveness for the Champions was assessed through change in vaccine knowledge, trust and confidence, and communication self-efficacy and for the community attendees through change in intention to vaccinate and support for vaccines. We did not measure vaccine uptake as the study was not controlled or powered to show a difference between an intervention and control group. Over the study period, HPV vaccine coverage increased by 12% [13,28] and the Ministry of Health believed the program had been well received in the community. However, the Ministry ran an HPV vaccine campaign in schools in August and September 2023 and as such, it is difficult to isolate the contribution made by the Vaccine Champions program. This would need to be explored in a randomised controlled trial

## Conclusions

Engaging diverse community leaders outside the health sector including teachers, local officials, nurses, and representatives from disability, youth, and sports groups to become vaccine advocates in their community can improve vaccine trust and confidence. This community engagement intervention showed impact at the co-design, Champions and community level according to the RE-AIM evaluation indicators. The study also identified important lessons that may be useful for other countries in the Pacific region and beyond to improve HPV vaccine uptake. Although HPV vaccine coverage increased in Tonga during the program period, there were other external influencing factors. We are planning hybrid cluster randomised controlled trials measuring vaccine coverage as well as implementation and cost outcomes in other countries in the region to build robust evidence of the program. The Vaccine Champions program is continuing in Tonga in 2025, with key adaptations to implementation and a focus on local ownership and sustainability.

## Supporting information

**S1 Table. Effectiveness of Vaccine Champions training.**
(DOCX)

**S2 Table. Characteristics of community attendees completing post-session survey (N = 269).**
(DOCX)

**S1 Checklist. Inclusivity in global research.**
(DOCX)

## Acknowledgments

The authors would like to thank the study participants for their valuable time and contributions, and Suzanna Vidmar for her data analysis support.

## Author contributions

**Conceptualization:** Yasmin Mohamed, Isabella Overmars, Ofakiokalani Tukia, Luisa Vodonaivalu, Reynold 'Ofanoa, Julie Leask, Holly Seale, Kylie Jenkins, Halitesh Datt, Sonya Sagan, Jessica Kaufman, Margie Danchin.

**Data curation:** Yasmin Mohamed, Isabella Overmars.

**Formal analysis:** Yasmin Mohamed, Isabella Overmars, Emma Luey, 'Asinate Toluta'u, Meleane Lomu, 'Ungatea Kata, Jessica Kaufman, Margie Danchin.

**Funding acquisition:** Julie Leask, Holly Seale, Jessica Kaufman, Margie Danchin.

**Investigation:** Yasmin Mohamed, Emma Luey, 'Asinate Toluta'u, Meleane Lomu, Luisa Vodonaivalu.

**Methodology:** Yasmin Mohamed, Isabella Overmars, Ofakiokalani Tukia, Emma Luey, 'Asinate Toluta'u, Meleane Lomu, Luisa Vodonaivalu, Afu Tei, Reynold 'Ofanoa, 'Ungatea Kata, Holly Seale, Kylie Jenkins, Kshitij Joshi, Halitesh Datt, Sonya Sagan, Michelle Dynes, Jessica Kaufman, Margie Danchin.

**Project administration:** Yasmin Mohamed, Isabella Overmars, Emma Luey, 'Asinate Toluta'u, Meleane Lomu, Luisa Vodonaivalu.

**Resources:** Ofakiokalani Tukia, Emma Luey, Afu Tei, 'Ungatea Kata, Julie Leask, Kshitij Joshi, Halitesh Datt, Sonya Sagan, Michelle Dynes.

**Supervision:** Ofakiokalani Tukia, Reynold 'Ofanoa, 'Ungatea Kata, Julie Leask, Holly Seale, Kylie Jenkins, Kshitij Joshi, Jessica Kaufman, Margie Danchin.

**Validation:** Yasmin Mohamed, Isabella Overmars, Emma Luey, Jessica Kaufman, Margie Danchin.

**Visualization:** Yasmin Mohamed, Isabella Overmars, Michelle Dynes, Jessica Kaufman, Margie Danchin.

**Writing – original draft:** Yasmin Mohamed.

**Writing – review & editing:** Isabella Overmars, Ofakiokalani Tukia, Emma Luey, 'Asinate Toluta'u, Meleane Lomu, Luisa Vodonaivalu, Afu Tei, Reynold 'Ofanoa, 'Ungatea Kata, Julie Leask, Holly Seale, Kylie Jenkins, Kshitij Joshi, Halitesh Datt, Sonya Sagan, Michelle Dynes, Jessica Kaufman, Margie Danchin.

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
