## [Decision Letter · Decision Letter 0]

27 Jun 2025

PGPH-D-25-00819

RE-AIM evaluation of a community-based vaccine education and communication program to improve human papillomavirus vaccine uptake in Tonga

Dear Dr. Mohamed,

Thank you for submitting your manuscript to PLOS Global Public Health. After careful consideration, we feel that it has merit but does not fully meet PLOS Global Public Health’s publication criteria as it currently stands. Therefore, we invite you to submit a revised version of the manuscript that addresses the points raised during the review process.

Please note that we have only been able to secure a single reviewer to assess your manuscript. We are issuing a decision on your manuscript at this point to prevent further delays in the evaluation of your manuscript. Please be aware that the editor who handles your revised manuscript might find it necessary to invite additional reviewers to assess this work once the revised manuscript is submitted. However, we will aim to proceed on the basis of this single review if possible. 

The reviewer has raised a number of concerns that need attention. They have mentioned several points to clarify in the methods and results sections. They also mention limitations that should be included and suggest adding a section on male vaccine champions. Additionally, they highlight a few sections of the manuscript that could have improved sentence structure and flow.  

Could you please revise the manuscript to carefully address the concerns raised?

We look forward to receiving your revised manuscript.

Kind regards,

Katherine Demi Kokkinias, Ph.D.

Staff Editor

Journal Requirements:

2. In the ethics statement in the Methods, you have specified that verbal consent was obtained. Please provide additional details regarding how this consent was documented and witnessed, and state whether this was approved by the IRB.

3. Please provide separate figure files in .tif or .eps format.

4. In the online submission form, you indicated that [The de-identified datasets used and/or analysed during the current study are available from the corresponding author on reasonable request.].

a. In a public repository,

b. Within the manuscript itself, or

c. Uploaded as supplementary information.

Additional Editor Comments (if provided):

Reviewers' comments:

Reviewer's Responses to Questions

**Comments to the Author**

1. Does this manuscript meet PLOS Global Public Health’s publication criteria?

Reviewer #1: Yes

2. Has the statistical analysis been performed appropriately and rigorously?

Reviewer #1: Yes

3. Have the authors made all data underlying the findings in their manuscript fully available (please refer to the Data Availability Statement at the start of the manuscript PDF file)?

Reviewer #1: No

4. Is the manuscript presented in an intelligible fashion and written in standard English?

Reviewer #1: Yes

Reviewer #1: Thank you for the opportunity to review this paper and learn about this program. HPV vaccination uptake is a critical public health issue, and this program is a feasibly and potentially highly effective intervention. I have a few comments and questions, primarily regarding the quantitative methods:

Abstract

-The word diverse is used twice in the sentence starting “Between March and December 2023” – consider more varied language

-Can you clarify when the program was actually run (dates), what data were collected, and who the participants were? The current abstract makes it sound as though the main focus was the co-design workshop.

Intro

-In the intro, there’s a focus on the intention-uptake gap, but this intervention seems to focus on intention. I recognize that you couldn’t measure uptake, but can you comment a bit on why intention and programs like the Vaccine Champion program, are still relevant?

Methods

-Is there a citation for roll out in Vietnam? (p5 line 92)

-Was any specific type of voting using in co-design workshop (ranked choice, consensus-building methods)?

-The data collection section is slightly confusing. It goes from a high level overview of the phases, to data collection among the vaccine champions, then finishes with the co-design participants. Can you reorganize so it follows the structure of the table and the progress of the study phase? I’d suggest [study phase], [participants], [data collection methods].

Results

-What were the results of the voting on program characteristics in the co-design-workshop?

-What are disability groups, in this context?

-What was the average size of the vaccine info sessions? Any variability by region/vaccine champion/other factors?

-Did you consider conducting any statistical analyses of your outcomes by sociodemographic characteristics?

-Did you consider the effect of clustering by vaccine champion (e.g., participants in sessions run by the same champion may be more similar to each other than participants in sessions run by a different vaccine champion)?

Discussion

-In the on-going program, will there be any additional compensation for vaccine champions as suggested by participants?

-Thank you for the discussion of gender-inclusive HPV vaccine policies. In Tonga, who typically holds decision-making power for medical decisions for children (mothers vs fathers)? Could male vaccine champions be useful in speaking with fathers?

-A key limitation is social desirability bias, both in how people respond to survey questions and who even responds in the first place. Can you discuss the impact of low participation in the post-event surveys on your results?

**Do you want your identity to be public for this peer review?** For information about this choice, including consent withdrawal, please see our Privacy Policy

Reviewer #1: No

---

## [Decision Letter · Decision Letter 1]

30 Sep 2025

PGPH-D-25-00819R1

RE-AIM evaluation of a community-based vaccine education and communication program to improve human papillomavirus vaccine uptake in Tonga

Dear Dr. Mohamed,

Thank you for submitting your manuscript to PLOS Global Public Health. After careful consideration, we feel that it has merit but does not fully meet PLOS Global Public Health’s publication criteria as it currently stands. Therefore, we invite you to submit a revised version of the manuscript that addresses the points raised during the review process.

We look forward to receiving your revised manuscript.

Kind regards,

Miquel Vall-llosera Camps

Staff Editor

Journal Requirements:

Additional Editor Comments:

We found necessary to invite an additional reviewer to assess your work. Please revise the manuscript to address all the reviewer's comments in a point-by-point response in order to ensure it is meeting the journal's publication criteria. Please note that the revised manuscript will need to undergo further review, we thus cannot at this point anticipate the outcome of the evaluation process.

Reviewers' comments:

Reviewer's Responses to Questions

**Comments to the Author**

Reviewer #1: All comments have been addressed

Reviewer #2: (No Response)

publication criteria?

Reviewer #1: Yes

Reviewer #2: Yes

3. Has the statistical analysis been performed appropriately and rigorously?

Reviewer #1: Yes

Reviewer #2: Yes

4. Have the authors made all data underlying the findings in their manuscript fully available (please refer to the Data Availability Statement at the start of the manuscript PDF file)?

Reviewer #1: Yes

Reviewer #2: No

5. Is the manuscript presented in an intelligible fashion and written in standard English?

Reviewer #1: Yes

Reviewer #2: Yes

Reviewer #1: Thank you for the thorough and thoughtful response to my comments. I believe this paper makes an important contribution to the literature on HPV vaccination uptake, and I appreciate that the level of detail you have provided makes it possible for others in different contexts to draw on your experiences when building their own programs.

Reviewer #2: The purpose of this study was to evaluate an HPV vaccine uptake campaign in Tonga. The manuscript could be strengthened by clarifying differences in evaluating program planning/adaptation, facilitator training, and implementation of the program itself.

The introduction focuses exclusively on cervical cancer, although HPV causes multiple types of cancers. The first paragraph also only mentions girls being vaccinated, reinforcing the feminization of HPV. Why is the HPV vaccine not available for boys in Tonga?

When the Vaccine Champions program was evaluated in Fiji and Vietnam – was this also in relation to COVID-19 or HPV vaccine? Or, was this the first time it had been used for HPV vaccine?

There could be more of a focus in the title/abstract/introduction that this program was also adapted, not just evaluated.

The introduction states that the purpose of this study was to evaluate the Vaccine Champions Program – but the methods and results also include a lot of program planning.

It is mentioned that an implementation science approach is used- more information needed about that.

More information about what co-design is would be helpful.

It seems that phase 1 and 2 are more about program planning than program evaluation

Phase 2 mentions training, but not what the training entailed. Was this the Vaccine Champions themselves being trained,?

Phase 3 mentions community attendees – are they attending the Vaccine Champions program or the co-design workshop?

A clearer differentiation between adapting the program using co-design, training the vaccine champions, and then evaluating the implementation of the program itself would strengthen the paper.

In Table 1 – it seems that you are evaluating both the development of the program, the training of the Vaccine champions, and the implementation of the program – is that correct? (For example, with co-design participants it says the RE-AIM domain of implementation was included, but it’s unclear what program was implemented during this co-design)

It is not clear how the vaccine champions were trained – were they given the same program that they would then be teaching to others? In table one it mentions “Implementation: adaptations made to program implementation” – what does this mean?

I am a little bit confused about RE-AIM being applied to the adaptation of the program and the training of the champions as well as to the program itself. I know that RE-AIM has been used as a process framework to plan implementation and adaptations – so is it being used here for both process and outcome evaluation? Clarifying that in the study design would be helpful. Currently it is worded in a way that suggests outcome evaluation only.

“We evaluated the three program phases using qualitative and quantitative methods (Table 1).” – would be helpful to break this down into what is program planning, implementation, process and outcome eval.

How many team members coded each interview? How were themes determined within each area of RE-AIM?

In the results, it is mentioned that “one participant felt…”. Does this represent a theme or did only one participant feel that way? Unclear how it was decided what quotes and views are included.

“Over the study period, HPV vaccine coverage increased by 12 % and the 480 Ministry of Health believed the program had been well received in the community.” – need a citation for this

**Do you want your identity to be public for this peer review?** For information about this choice, including consent withdrawal, please see our Privacy Policy

Reviewer #1: No

Reviewer #2: No

---

## [Decision Letter · Decision Letter 2]

28 Oct 2025

RE-AIM evaluation of a community-based vaccine education and communication program to improve human papillomavirus vaccine uptake in Tonga

PGPH-D-25-00819R2

Dear Ms Mohamed,

We are pleased to inform you that your manuscript 'RE-AIM evaluation of a community-based vaccine education and communication program to improve human papillomavirus vaccine uptake in Tonga' has been provisionally accepted for publication in PLOS Global Public Health.

Best regards,

Julia Robinson

Executive Editor

Reviewer Comments (if any, and for reference):

Reviewer's Responses to Questions

**Comments to the Author**

Reviewer #1: All comments have been addressed

Reviewer #2: All comments have been addressed

publication criteria?

Reviewer #1: Yes

Reviewer #2: Yes

3. Has the statistical analysis been performed appropriately and rigorously?

Reviewer #1: Yes

Reviewer #2: Yes

4. Have the authors made all data underlying the findings in their manuscript fully available (please refer to the Data Availability Statement at the start of the manuscript PDF file)?

Reviewer #1: Yes

Reviewer #2: Yes

5. Is the manuscript presented in an intelligible fashion and written in standard English?

Reviewer #1: Yes

Reviewer #2: Yes

Reviewer #1: (No Response)

Reviewer #2: The authors addressed my commments.

**Do you want your identity to be public for this peer review?** For information about this choice, including consent withdrawal, please see our Privacy Policy

Reviewer #1: No

Reviewer #2: No
